# Knowledge, Attitudes, and Beliefs Regarding Drug Abuse and Misuse among Community Pharmacists in Saudi Arabia

**DOI:** 10.3390/ijerph17041334

**Published:** 2020-02-19

**Authors:** Abdulmajeed M. Mobrad, Sultan Alghadeer, Wajid Syed, Mohamed N. Al-Arifi, Arafah Azher, Mansour S Almetawazi, Salmeen D. Babelghaith

**Affiliations:** 1Prince Sultan College for EMS, King Saud University, Riyadh 12642, Saudi Arabia; amobrad@ksu.edu.sa; 2Department of Clinical Pharmacy, College of Pharmacy, King Saud University, Riyadh 11451, Saudi Arabia; salghadeer@ksu.edu.sa (S.A.); malarifi@ksu.edu.sa (M.N.A.-A.); aazhar@ksu.edu.sa (A.A.); mansour@ksu.edu.sa (M.S.A.); Sbabelghaith@ksu.edu.sa (S.D.B.)

**Keywords:** drug abuse, knowledge, attitude, opinion, misuse, community pharmacy, Saudi Arabia

## Abstract

*Background and Objective:* Several over-the-counter drugs have been documented as an essential issue in the community pharmacy setting owing to their liability to abuse. Pharmacists act as a critical monitor for these concerns and evaluate the seriousness of the patients’ condition. Therefore, this study aimed to assess the knowledge, attitudes, and beliefs regarding drug abuse and misuse among pharmacists at a community pharmacy in Riyadh city, Saudi Arabia. *Methods:* A cross-sectional study using a validated self-administered questionnaire was carried out among community pharmacists over three months April to June 2019. The survey had 25 items on the experience, knowledge, attitudes, beliefs, strategies, and opinions of participants toward drug abuse and misuse. *Results:* A total of 239 community pharmacists responded to the survey. About 84% of them had received training on drug misuse or abuse. The majority of community pharmacists (85.8%) would like to be provided educational programs on drug abuse in the future. Nearly all the pharmacists (94.9%) reported providing suitable advice to suspected drug misusers either in written or oral form at their pharmacies. Approximately 31% agreed or strongly agreed to dispense controlled drugs through a pharmacy. Regarding the ethical matter of selling misusers controlled drugs, 93.7% of the respondents believed that it is deceptive to offer misusers controlled medications. A comparison of knowledge and beliefs (univariate analysis) showed that the results were significant only for respondents who had graduated from Yemen (*p* = 0.007) and respondents who had an experience of four to six years or more (*p* < 0.01). *Conclusion:* The findings revealed that the majority of community pharmacists had been trained in recognizing drug abuse or dependence during their pharmacy college education. In addition, majority of them reported that they warned or counseled patients about the occurrence of adverse drug reactions to specific medications. However, majority of them agreed that selling controlled drugs is unethical in a community pharmacy. Thus, effective implementation of pharmaceutical rules and laws is a fundamental need in the Saudi Arabian health care system and we suggest stringent execution of the regulations by the Saudi health care authorities.

## 1. Introduction

Several over-the-counter (OTC) drugs such as opioids, antihistamines, and laxatives have been documented as an essential issue in the community pharmacy setting owing to their liability to abuse [1,2,3,4,5]. OTC drugs may be associated with abuse and misuse problems, which are considered inappropriate clinically and ethically. Abuse of OTC drugs refers to utilizing a drug for nonmedical or unintended purposes, such as to achieve mind-altering effects or lose weight [1,6]. Misuse of OTC drugs refers to using medications for therapeutic purposes but in a wrong manner, typically in terms of dosage or period of use [1].

OTC drug use in Saudi Arabia is reportedly high [7]. Some studies have suggested that the prevalence rates of substance abuse in Arab countries are accelerating quickly as a result of the extraordinarily fast economic development [8,9]. In Saudi Arabia, a regional study revealed that psychoactive substance consumption was slightly high, a reality illustrated by transversal epidemiologic investigations carried out at primary school, secondary school, and advanced education establishments [9]. The most used illegal psychoactive substances were cannabis (51.4%), followed by glue/solvents (48.6%) and amphetamine (45.7%). Another regional study found that 10% to 79.6% of users of central nervous system stimulant drugs met the criteria for substance dependency [10]. Narcotic and controlled drugs are prescribed based on specialty, department (emergency vs. clinic), or supply of prescribed medication according to the policy and legislation for delivery of narcotic and controlled medications released by the Ministry of Health in Saudi Arabia. For instance, prescribing narcotic drugs for more than 60 days’ supply is consider a violation of the above-mentioned legislation [11].

Pharmacists have been considered as observers in drug abuse [12]. Pharmacists act as an essential monitor for abuse and misuse concerns, evaluate seriousness of the patients’ condition, allude patients to suitable levels of care, as well as offer proper directions to those in recuperation [13]. Despite this crucial role, many pharmacists might have inadequate knowledge about drug abuse. Pharmacists may be poorly trained to recognize, interfere with, or manage patients with drug abuse problems. A study conducted in 2005 showed that only 48% of pharmacists have been trained in preventing drug diversion [13,14]. Insufficient education, namely regarding early documentation, referral, and patient counseling, was reported as obstacles to effective pharmacist mediation with the drug abuser [15]. Many studies of drug abuse training courses in colleges of pharmacy demonstrated the requirement for additional training of pharmacists for counseling drug-dependent patients [16,17]. Improving the knowledge of drug misuse/abuse among community pharmacists (CPs) by implementing useful programs will enhance their effectiveness in limiting OTC drug misuse and abuse. This study aimed to evaluate CPs’ knowledge, attitudes, and beliefs regarding drug misuse and abuse in Saudi Arabia.

## 2. Methods

### 2.1. Design and Setting 

A descriptive cross-sectional study utilizing an approved questionnaire [6] was conducted among CPs in the capital city of Saudi Arabia from April to June 2019 to evaluate their knowledge, attitudes, and beliefs regarding drug abuse and misuse. The estimated sample size of this study was based on a previously published study in the same area that mentioned 2000 pharmacists were listed with the health affairs directorate in Riyadh city [16]. To calculate the sample size, we expected half of the CPs to have substantial information about drug abuse, and a sum of 350 CPs would furnish a sample size with 5% margin of error and 95% confidence level [17].

The utilized questionnaire was composed of 25 items across six sections: (1) demographic information (five binary questions with yes/no response), (2) experience of patients’ drug abuse and misuse (four binary questions with yes/no response), (3) knowledge and attitudes regarding abuse and misuse of medications (three binary questions with yes/no response), (4) beliefs regarding abuse and misuse of drugs (four questions rated on a five-point scale ranging from strongly agree to not sure), (5) suggested strategies to reduce drug misuse and abuse (three binary questions with yes/no response), and (6) views on strategies used to decrease drug abuse and misuse (six questions rated on a five-point scale ranging from strongly agree to not sure).

A pilot study was conducted among CPs under the supervision of an investigator to evaluate the responses to measure the validity of the questionnaire to understand the best way for data collection and management. The pilot study was completed in two weeks and involved 20 subjects. The reliability value was 0.76. All necessary additions or changes were made to the study tools. The results of the pilot study have not been included in the central review. 

The study was granted necessary ethical permission from the King Saud University College of Medicine (approval number E-19-3983). All the CPs were informed that the survey would be anonymous and that the recorded responses would be used for scientific purposes only.

### 2.2. Data Analysis

Descriptive statistics involved proportions; means and frequency distribution were calculated for each variable. Statistical package for social sciences version 24.0 (SPSS Inc., Chicago, IL, USA) was used for computations. Multivariate and univariate analyses were carried out to find any association between the variables; a *p*-value of <0.5 were considered statistically significant.

## 3. Results

A total of 239 CPs responded to the survey (response rate of 68.2%). All the respondents were male, and about half of them were aged 30–40 years. The majority of CPs (92.4%) were pharmacy technicians. Around 75% of CPs had graduated from Egypt, and 53% had more than six years of experience. Table 1 represents the demographic data of the respondents. 

In total, 57.7% of CPs reported that their family members or friends had experienced drug misuse problems. About 84% of respondents had received training about drug misuse or abuse since graduation from pharmacy school. Most CPs (85.8%) said they would appreciate receiving educational programs about drug abuse in the future, as shown in Table 2. 

When the CPs were asked about the distinction between the terms “drug abuse” and “misuse,” 90% of them knew the difference. The majority of respondents (94.9%) reported providing suitable advice to suspected drug misusers in written or oral form at their pharmacies on the hazards or on how to treat the hazards. When CPs were asked about dispensing controlled drugs through the pharmacy rather than a central clinic, 59% of CPs agreed or strongly agreed to dispensing controlled drugs through pharmacy, whereas 41% of CPs completely disagreed. Regarding the ethical concern in selling misused controlled drugs, 93.7% of the CPs thought that it is untrustworthy to offer misusers controlled medications, as shown in Table 3.

As regards the suggested strategies to decrease drug misuse/abuse in community pharmacies, 94.6% of the CPs reported that they warned or counseled patients about the likely adverse drug reactions (ADR) to specific medications. However, 83.7% said that they proposed more secure options to abused drugs (Table 4). 

A high number of CPs agreed with the recommendation that all CPs must be prepared for perceiving drug abusers (93.3%), educated about the medications abused in the neighborhood (96.3%), and educated on ways of dealing with drug abusers (90.8%). In total, 91.2% of the respondents recommended that all supposed abusers should be referred to a medical doctor (Table 5).

A comparison of knowledge and beliefs (univariate analysis) showed that the results were significant only for respondents who had graduated from Yemen (*p* = 0.007), and who had four to six years’ experience or more (*p* < 0.01; see Table 6). 

A logistic regression model adjusted for country (Yemen) and years of experience for exploring in-depth knowledge showed that CPs from Yemen had significant in-depth knowledge (*p* = 0.028, OR = 0.088; see Table 7).

## 4. Discussion

Several studies have reported that pharmacists have insufficient training on drug abuse. More specifically, pharmacy students and pharmacists are inadequately educated to classify, intervene with, or treat patients and collaborators with substance abuse problems [14,17]. A study showed that only 71% of CPs reported receiving formal education about substance abuse since graduation from pharmacy school [15]. In another study carried out by the Center on Addiction and Substance Abuse in 2005, only 48% (*n* = 1030 registered pharmacists) stated that they had received any training in avoiding drug diversion [13]. However, the results of our study suggested that the majority of CPs had been trained about identifying abuse or dependence during their pharmacy bachelor’s degree. Despite this fact, around 85.8% of CPs were interested in receiving future educational programs about drug abuse. These findings demonstrate a critical emphasis on performing pre- and post-graduate substance abuse education and training courses and programs. Training or specialty in pharmacy residency programs should be provided, and pharmacist involvement in community service and drug abuse management research should be encouraged [14].

In community pharmacy settings, the pharmacist spends more time with patients for personal counseling. Therefore, they are capable of monitoring abuse problems, evaluate their severity, and offer proper counseling for individuals in recovery. Our study revealed that the majority of respondents provided appropriate advice to suspected drug abusers, or they attempted to treat the risks of their behavior. These findings demonstrate that pharmacists should be trained about drug abuse and hold an ethic that supports the necessary placement of such skills. CPs need logistical support in their practice setting to offer such counseling and services; therefore, pharmacy owners and managers must be convinced about the importance of drug abuse management services, and CPs should be provided adequate time to involve patients in the counseling area that gives confidentiality to conduct the service effectively in a professional manner [14].

Although the license laws and ethics in the pharmacy profession aim to protect the community and establish the boundaries of professional conduct [14], our study found that about 36.4% of CPs believed that administrating controlled medications without a prescription is a necessary foundation of income for the pharmacy. CPs often maintain a business together with their health care services, which may prompt ethical issues. Al-Mohamadi and his colleagues conducted a study to determine the dispensing behavior of pharmacists in retail pharmacy practice and to measure their attitude toward dispensing non-OTC drugs. They reported that 89.5% of CPs allocated antipsychotics just by following coworkers’ request without seeking medical recipes [8]. In this study, 38.9% of CPs stated that they transacted with supposed drug misusers in the same manner as with other clients.

These results confirm previous findings that reported the absence of Saudi Food and Drug Authority (SFDA) regulations for controlling the dispensing patterns in the Saudi community pharmacy. The SFDA manages all matters associated with pharmaceutical products, involving prescription medications, behind-the-counter products, OTC products, herbal products, and food supplements. However, it does not control medical or pharmacy services. Nevertheless, the SFDA can place prescription restrictions based on specialty and license classification of physicians [8,18]. Approximately 13.59% of CPs reported complete absence of the SFDA in controlling or monitoring their services [8].

The majority of respondents believed that they could reduce misuse and abuse of drugs in the community through different strategies, including advising patients about expected ADR to specific medications and recommending safer substitutes for the abused drug. These findings support the ethical goals in a pharmacist’s profession. Pharmacists should carry the responsibility of drug abuse prevention and education efforts. CPs should be aware of the practice policy and also asked to be part of the care plan for consumers receiving drugs of dependence [19,20]. There are numerous activities that pharmacists can undertake to avert drug abuse. Pharmacists can assist with dependence or abuse diagnosis, initiate interview tools, make a list of plans for dependence management in their region or workplace, launch a public support group for patients with a particular disease, intervene directly with these patients through motivational meetings, contribute to multidisciplinary health teams assisting these patients, and share data with their families about medications and their particular unsafe effects [14].

This study has some limitations. The sample size was small, and the research was confined to Riyadh city. Therefore, the outcomes of the current study can only represent the situation in Riyadh city. In addition, to address sampling bias due to respondents who did not respond to the survey, it is recommended that future studies investigate the factors or reasons that prevented them from participating.

## 5. Conclusions

The findings of this study revealed that the majority of CPs had been trained in recognizing abuse or dependence of drugs during their pharmacy college education and they could provide appropriate advice to suspected drug abusers. In addition, most of them reported that they warned or counseled patients about the occurrence of ADR to specific medications. However, the majority of CPs agreed that selling controlled drugs in a community pharmacy is unethical. Thus, effective enforcement of the pharmaceutical rules and laws by the Saudi health care authorities is considered as an essential demand in the Saudi Arabian health system.

## Figures and Tables

**Table 1 ijerph-17-01334-t001:** Demographic information of the participants (*n* = 239).

Demographic	Description (*n* = 239)
Age group	
24–30	74 (31.1%)
31–40	131 (55%)
41–50	27 (11.3%)
more than 51	6 (2.5%)
Qualifications	
master’s degree	5 (2.1%)
Pharm D	13 (5.5%)
Pharmacy technician	219 (92.4%)
Graduation country	
Saudi Arabia	3 (2.1%)
Egypt	108 (75%)
Yemen	8 (5.6%)
Pakistan	3 (2.1%)
India	4 (2.8%)
Others	18 (12.5%)
Years of experience	
1–2 years	11 (4.6%)
2–4 years	49 (20.5%)
4–6 years	53 (22.2%)
more than 6 years	126 (52.7%)

**Table 2 ijerph-17-01334-t002:** Experience of participants regarding the problems of drug abuse and misuse (*n* = 239).

Variable	Description (*n* = 239)
Do you have any friends or relatives who faced problems in using drugs?	
Yes	138 (57.7%)
No	101 (42.3%)
Have you ever received training about drug misuse at the school of pharmacy?	
yes	201 (84.1%)
no	38 (15.9%)
Have you had any course about drug abuse during your study?	
yes	202 (84.5%)
no	37 (15.5%)
Do you like to receive training on drug misuse in the future?	
yes	205 (85.8%)
no	34 (14.2%)

**Table 3 ijerph-17-01334-t003:** Community pharmacists’ knowledge and attitudes and believes about misuse and abuse of drugs (*n* = 239).

Variables	*N* (%)
Do you know the difference between the terms ‘drug abuse’ and ‘misuse’?	
Yes	215 (90.0)
No	24 (10)
It is suitable to advise suspected drug misuses (in a written or verbal form) in a pharmacy about the risks or the way of treating the risks of their behavior.	
Strongly agree	138 (57.7)
Agree	89 (37.2)
Strongly disagree	3 (1.3)
Disagree	5 (2.1)
Not Sure	4 (1.7)
I am willing to advise suspected drug misuse (in a written or verbal form) about the risk or the way of treating the risks of their behavior.	
Strongly agree	133 (55.6)
Agree	89 (37.2)
Strongly disagree	6 (2.5)
Disagree	5 (2.1)
Not Sure	6 (2.5)
I believe that controlled drugs must be dispensed through the pharmacy instead of a central clinic.	
Strongly agree	74 (31)
Agree	67 (28)
Strongly disagree	32 (13.4)
Disagree	56 (23.4)
Not Sure	10 (4.2)
I believe that it is unethical to sell misuses controlled drugs.	
Strongly agree	160 (66.9)
Agree	64 (26.8)
Strongly disagree	4 (1.7)
Disagree	9 (3.8)
Not Sure	2 (0.8)
Dispensing controlled dugs without a medical recipe is an important source of money for the pharmacy.	
Strongly agree	31 (13)
Agree	56 (23.4)
Strongly disagree	94 (39.3)
Disagree	49 (20.5)
Not Sure	9 (3.8)
I deal with suspected drug misuses the same way as I deal with other customers.	
Strongly agree	45 (18.8)
Agree	48 (20.1)
Strongly disagree	53 (22.2)
Disagree	75 (31.4)
Not Sure	18 (7.5)

**Table 4 ijerph-17-01334-t004:** Proposed approaches by participants to lessen drug misuse and abuse in community pharmacy (*n* = 239).

Variable	*N* (%)
I warn/advise patients about the expected ADR associated with certain drugs.	
Yes	226 (94.6)
No	13 (5.4)
I suggest safer alternatives for the abused drug.	
Yes	200 (83.7)
No	39 (16.3)
Do you call nearby pharmacies to inform them about a suspected abuser?	
Yes	144 (60.3)
No	95 (39.7)

**Table 5 ijerph-17-01334-t005:** Views of CPs on approaches used to decrease drug misuse and abuse (*n* = 239).

Variable	*N* (%)
How would you rate the level of control over drugs by official authorities in the pharmacy you have been trained or working in?	
Strongly controlled	140 (58.6)
Controlled to some extent	95 (39.7)
Badly controlled	4 (1.7)
All pharmacy staff must be trained on recognizing drug abusers.	
Strongly agree	142 (59.4)
Agree	81 (33.9)
Strongly disagree	1 (0.4)
Disagree	5 (2.1)
Not Sure	10 (4.2)
All pharmacy staff must be informed of the kinds of drugs abused in the local area of the pharmacy.	
Strongly agree	139 (58.2)
Agree	91 (38.1)
Strongly disagree	5 (2.1)
Disagree	3 (1.3)
Not Sure	1 (0.4)
All staff should be trained on methods of dealing with drug abusers.	
Strongly agree	151 (63.2)
Agree	66 (27.6)
Strongly disagree	3 (1.3)
Disagree	10 (4.2)
Not Sure	9 (3.8)
All staff should be informed of drugs liable to abuse.	
Strongly agree	144 (60.3)
Agree	84 (35.1)
Strongly disagree	6 (2.5)
Disagree	2 (0.8)
Not Sure	3 (1.3)
All suspected abusers should be referred to a physician.	
Strongly agree	140 (58.6)
Agree	78 (32.6)
Strongly disagree	5 (2.1)
Disagree	6 (2.5)
Not Sure	10 (4.2)
Knowledge & belief	
High	151 (63.2)
Low	88 (36.8)

**Table 6 ijerph-17-01334-t006:** Comparison regarding Knowledge & belief (univariate analysis).

Characters	Knowledge & Belief		*p* Value *
	High (*n* = 151, 63.2%)	Low (*n* = 88, 36.8%)	
Age group			
24–30	50 (33.3)	24 (27.3)	0.330
31–40	80 (53.3)	51 (58)	0.489
41–50	14 (9.3)	13 (14.8)	0.201
more than 51	6 (4)	0 (0)	0.087
Qualifications			
Master’s degree	3 (2)	2 (2.3)	1.000
Pharm D	8 (5.4)	5 (5.7)	1.000
Pharmacy technician	138 (92.6)	81 (92)	0.872
Graduation country			
Saudi Arabia	2 (2.3)	1 (1.8)	1.000
Egypt	67 (77)	41 (71.9)	0.491
Yemen	1 (1.1)	7 (12.3)	0.007
Pakistan	2 (2.3)	1 (1.8)	1.000
India	3 (3.4)	1 (1.8)	1.000
Others	12 (13.8)	6 (10.5)	0.616
Years of experience			
1–2 years	7 (4.6)	4 (4.5)	1.000
2–4 years	35 (23.2)	14 (15.9)	0.179
4–6 years	44 (29.1)	9 (10.2)	0.001
more than 6 years	65 (43)	61 (69.3)	0.000
Do you have any friends or relatives who faced problems in using drugs?			
Yes	95 (62.9)	43 (48.9)	0.034
No	56 (37.1)	45 (51.1)	
Have you ever received training about drug misuse at the school of pharmacy?			
yes	128 (84.8)	73 (83)	0.712
no	23 (15.2)	15 (17)	
Have you had any course about drug abuse during your study?			
yes	128 (84.8)	74 (84.1)	0.889
no	23 (15.2)	14 (15.9)	
Do you like to receive training on drug misuse in the future?			
yes	128 (84.8)	77 (87.5)	0.560
no	23 (15.2)	11 (12.5)	

* Chi square test.

**Table 7 ijerph-17-01334-t007:** Logistic regression model to explore predictors of Knowledge (Multivariate analysis).

Characteristics	*p* Value	OR	95% CI for OR		
Yemen country	0.028	0.088	0.010	-	0.774
Years of experience					
1–2 years	……	Reference			
2–4 years	0.454	2.093	0.303	-	14.464
4–6 years	0.509	1.924	0.276	-	13.409
more than 6 years	0.546	0.579	0.098	-	3.408
Having any friends or relatives who faced problems in using drugs	0.384	1.383	0.667	-	2.867

OR = odds ratio, CI = confidence interval. A Logistic regression model adjusted for Yemen country, and years of experience for exploring the in depth knowledge and showed that Yemen country had a significant deep knowledge (*p* = 0.028, OR = 0.088). From 2 to 4 years of experience had the highest odds (OR = 2.093, CI = 0.303–14.464). Participants who had more than six years of experience had the lowest odds (OR = 0.579, CI = 0.098–3.408).

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
