# Peer review of "Knowledge, Attitudes, and Beliefs Regarding Drug Abuse and Misuse among Community Pharmacists in Saudi Arabia"

_ijerph, 2020, doi:10.3390/ijerph17041334_

Round 1
Reviewer 1 Report
The manuscript needs significant background explanation to enable the reader to understand the context of the study. Therefore, it is suggested to present the current medicines supply legislation and regulations in Saudi Arabia to highlight why "it is deceptive to offer misusers controlled medications". The manuscript contains many factual inaccuracies e.g. "Approximately 33.4% were agreed or strongly agreed to dispense controlled drugs through pharmacy". To our knowledge, pharmacists dispense controlled drugs against a prescription. This is part of the pharmacy contract and is not subjected to additional training or personal views.
Grammar needs to be significantly improved as it is affecting the meaning overall in the manuscript. For example, “essential issue in the community pharmacy setting due to its liability to abuse”, “33.4% were agreed..”.
Inappropriate choice of words such as “Pharmacists have been considered as the 'custodians' of drug abuse”.
Some conclusions are made based on subjective answers from the participants. For example, "nearly all the pharmacists (94.7%) provided suitable advice to suspect drug misusers either in a written or oral form in their pharmacies". It is not clear how the provided advice was deemed suitable if this was based on a binary yes/no answer to a question.
Author Response
Dear editor in chief very good evening , thank you for the giving a chance to publish with you , we will be glad to publish , and please accpet the changes in mansucript ,
my kind request to accept the 2 authors as i mentioned in attached corrected mansucript . as both of them did regressiona and reviewed the file
i beg you to accept .
Reviewer 1 comments
Comments and Suggestions for Authors
The manuscript needs significant background explanation to enable the reader to understand the context of the study. Therefore, it is suggested to present the current medicines supply legislation and regulations in Saudi Arabia to highlight why "it is deceptive to offer misusers controlled medications". The manuscript contains many factual inaccuracies e.g. "Approximately 33.4% were agreed or strongly agreed to dispense controlled drugs through pharmacy". To our knowledge, pharmacists dispense controlled drugs against a prescription. This is part of the pharmacy contract and is not subjected to additional training or personal views.
We have done changes in manuscript.
Grammar needs to be significantly improved as it is affecting the meaning overall in the manuscript. For example, “essential issue in the community pharmacy setting due to its liability to abuse”, “33.4% were agreed.”.
Approximately 31% were agreed or strongly agreed to dispense controlled drugs through a pharmacy (abstract section)
Inappropriate choice of words such as “Pharmacists have been considered as the 'custodians' of drug abuse”.
Pharmacists have been considered as the observer to drug abuse (introduction last paragraph)
Some conclusions are made based on subjective answers from the participants. For example, "nearly all the pharmacists (94.7%) provided suitable advice to suspect drug misusers either in a written or oral form in their pharmacies". It is not clear how the provided advice was deemed suitable if this was based on a binary yes/no answer to a question.
It was removed from the conclusion
Reviewer 2 Report
In the manuscript, Al-Arifi et al. performed a cross-sectional study to evaluate the knowledge, attitudes and opinions of drug abuse and misuse at community pharmacy in the capital city of Saudi Arabia from April 2019 to June 2019. Responses were received from 239 community pharmacists. According to the responses, they find many community pharmacists claimed that they had the training about drug misuse and drug abuse, and would like to provide advice. They also revealed that some pharmacists considered administrating controlled medications without a prescription was a substantial income for the pharmacy. However, this paper has several major flaws, which makes it not suitable to be accepted for publication.
The English language in the text is very hard to follow, as well as many grammar mistakes (such as “abuse use”, “239 CPs were responded to the survey”, and many tense errors, etc.). It is recommended that the author either ask a colleague whose native language is English to review the manuscript or use language editing services. It will impede the readers to understand what the author is really trying to convey.
All the data are descriptive with no further analysis. There are no statistical comparison to show if there is any significant difference between groups to a certain questions. Moreover, the author may consider using logistic regression model to explore the in depth knowledge.
According to the paper, around 40% of the recipients did not respond the survey, which is a big proportion. The author should provide the possible reason for the low response rate. It is possible that those who decided not to answer the survey were driven by certain factors, which would lead to the sampling bias.
In table 3, the question “Do you know the difference between the terms ‘drug abuse’ and ‘misuse’?” may not objectively reflect whether a pharmacist really understand the difference. Other questions should be included in the survey.
As the author mentioned, the small sample size and the single location where the survey was performed were limitations of the manuscript. However, in the conclusion, the author believed it is urgent to implement the pharmaceutical rules and laws. Since no evidence from other sides and other populations, the conclusion is not no solid.
Some numbers in all tables needs to be aligned and reformatted.
Author Response
Dear editor in cheif thank you for the comments we have given point to point explanation mainly major changes done in manuscript , please accept the changes in authors list as the both authors did regression and reviewed the Manuscript .
i sincearly aplogies for this issue and hoping to get published our research in your estmeed journal
In the manuscript, Al-Arifi et al. performed a cross-sectional study to evaluate the knowledge, attitudes and opinions of drug abuse and misuse at community pharmacy in the capital city of Saudi Arabia from April 2019 to June 2019. Responses were received from 239 community pharmacists. According to the responses, they find many community pharmacists claimed that they had the training about drug misuse and drug abuse, and would like to provide advice. They also revealed that some pharmacists considered administrating controlled medications without a prescription was a substantial income for the pharmacy. However, this paper has several major flaws, which makes it not suitable to be accepted for publication.
My apologies for the error we have made it correct in the conclusion section. This sentence was thought by our authors we corrected the sentence.
The findings of this study revealed that the majority of CPs have been trained in recognizing abuse or dependence of drugs during their studying at their pharmacy colleges and they can provide appropriate advice to the suspected drug abusers. However, majority of the CPs agreed that selling controlled drugs is unethical in community pharmacy. In addition, majority of the CPs reported that they warn or counsel patients about the occurrence of adverse drug reactions (ADR) related with specific medications. Therefore, we suggest further more implementation of the pharmaceutical rules and laws by the Saudi health care authorities and is considered as an essential demand in Saudi Arabian health care system.
The English language in the text is very hard to follow, as well as many grammar mistakes (such as “abuse use”, “239 CPs were responded to the survey”, and many tense errors, etc.). It is recommended that the author either ask a colleague whose native language is English to review the manuscript or use language editing services. It will impede the readers to understand what the author is really trying to convey.
My apologies for the error we have corrected all the grammar
All the data are descriptive with no further analysis. There is no statistical comparison to show if there is any significant difference between groups to a certain question. Moreover, the author may consider using logistic regression model to explore the in depth knowledge.
Thank you very much for the advice and we have done regression for the results
A comparison regarding Knowledge & belief (univariate analysis) showed that the only significance was shown between respondents who were graduated from Yemen (p=0.007), and respondents who had experience from four to six years or more (p < 0.01) (See Table. 6): A Logistic regression model adjusted for Yemen country, and years of experience for exploring the in depth knowledge and showed that Yemen country had a significant deep knowledge (p= 0.028, OR= 0.088). (See the table -7)
According to the paper, around 40% of the recipients did not respond the survey, which is a big proportion. The author should provide the possible reason for the low response rate. It is possible that those who decided not to answer the survey were driven by certain factors, which would lead to the sampling bias.Thank you for the comment. the major reason for this is the most of the respondents given incomplete answers, when we ask about this reason they said lack of time.
In table 3, the question “Do you know the difference between the terms ‘drug abuse’ and ‘misuse’?” may not objectively reflect whether a pharmacist really understand the difference. Other questions should be included in the surve
My apologies for the error, already we finfish the research so it will need the time to again start to collecting the data.
As the author mentioned, the small sample size and the single location where the survey was performed were limitations of the manuscript. However, in the conclusion, the author believed it is urgent to implement the pharmaceutical rules and laws. Since no evidence from other sides and other populations, the conclusion is not no solid.
We have made changes as per the comments. My apologies for the error. you can look in the conclusion.
Round 2
Reviewer 1 Report
Thanks for the authors for improving the manuscript.
The manuscript however requires substantial grammar edits and is still very difficult to follow.
Author Response
My apogloigies for the erros , we corrected for the english language using editage service , am here attaching the corrected file , as well as letter from editor
and we requested in first revision author changes , but still one author missing from the file what we requested. please kindly do need full
Reviewer 2 Report
Manuscript of "Knowledge, Attitude, and Belief of Drug Abuse and Misuse at Community Pharmacy Settings in Saudi Arabia " has been improved substantially over an earlier version. However, there are still some minor grammar errors, such as page 1, paragraph 2. "Some studies have been
suggested that the" should be changed to "Some studies suggested that the". A careful grammar check is needed before publication.
Author Response
My apologies for the errors , as per the reviewrs comments we submitted for grammer check and language using editage services
here am attahching the corrected file
please do needfull , as i requested author change in the first revision using author change form , but still the original first author missing from the list , please follow the below authors as well as the authors i mentioned in the main file ,
we are having so many papers to publish , and we will conitnue to publish with your journal , please od more and more support , as we rae gruop of researchers from the college of pharmacy . we badly need your full support and help to get publish
the corrected author list
Abdulmajeed M Mobrad 1 , Sultan Alghadeer 2 , Wajid syed 2 , Mohamed N Al Arifi 2 , Arafah A2 , Mansour S Almetawazi 2 , And Salmeen D Babelghaith2
1 Prince Sultan college for EMS , King Saud university , Riyadh Saud Arabia
2 Department of clinical pharmacy, college of pharmacy, king Saud university , Riyadh , Saud Arabia
